# Elemental Fingerprinting of Wild and Farmed Fish Muscle to Authenticate and Validate Production Method

**DOI:** 10.3390/foods11193081

**Published:** 2022-10-04

**Authors:** Renato Mamede, Irina A. Duarte, Isabel Caçador, Patrick Reis-Santos, Rita P. Vasconcelos, Carla Gameiro, Paula Canada, Pedro Ré, Susanne E. Tanner, Vanessa F. Fonseca, Bernardo Duarte

**Affiliations:** 1MARE—Marine and Environmental Sciences Centre & ARNET—Aquatic Research Infrastructure Network Associated Laboratory, Faculdade de Ciências da Universidade de Lisboa, Campo Grande, 1749-016 Lisboa, Portugal; 2Departamento de Biologia Vegetal da Faculdade de Ciências da Universidade de Lisboa, Campo Grande, 1749-016 Lisboa, Portugal; 3Southern Seas Ecology Laboratories, School of Biological Sciences, The University of Adelaide, Adelaide, SA 5005, Australia; 4IPMA—Instituto Português do Mar e da Atmosfera, Av. Dr. Alfredo Magalhães Ramalho 6, 1495-165 Algés, Portugal; 5Oceanic Observatory of Madeira, ARDITI, Madeira Tecnopolo, 9020-105 Funchal, Portugal; 6CIIMAR—Interdisciplinary Center of Marine and Environmental Research, University of Porto, 4450-208 Matosinhos, Portugal; 7MARE—Marine and Environmental Sciences Centre & ARNET—Aquatic Research Infrastructure Network Associated Laboratory, Laboratório Marítimo da Guia, Faculdade de Ciências, Universidade de Lisboa, Avenida Nossa Senhora do Cabo, 2750-374 Cascais, Portugal; 8Departamento de Biologia Animal da Faculdade de Ciências da Universidade de Lisboa, Campo Grande, 1749-016 Lisboa, Portugal

**Keywords:** fish, traceability, trace metals, seafood, aquaculture, TXRF

## Abstract

In the context of expanding fish production and complex distribution chains, traceability, provenance and food safety tools are becoming increasingly important. Here, we compare the elemental fingerprints of gilthead seabream (*Sparus aurata*) muscle from wild and different aquaculture productions (semi-intensive earth ponds and intensive sea cages from two locations) to confirm their origin and evaluate the concentrations of elements with regulatory thresholds (Cu, Hg, Pb and Zn). Using a chemometric approach based on multi-elemental signatures, the sample origin was determined with an overall accuracy of 90%. Furthermore, in a model built to replicate a real-case scenario where it would be necessary to trace the production method of *S. aurata* without reliable information about its harvesting location, 27 of the 30 samples were correctly allocated to their original production method (sea-cage aquaculture), despite being from another location. The concentrations of the regulated elements ranged as follows: Cu (0.140–1.139 mg/Kg), Hg (0–0.506 mg/Kg), Pb (0–2.703 mg/Kg) and Zn (6.502–18.807 mg/Kg), with only Pb presenting concentrations consistently above the recommended limit for human consumption. The present findings contribute to establishing elemental fingerprinting as a reliable tool to trace fish production methods and underpin seafood authentication.

## 1. Introduction

The global fish trade has been steadily growing over the years, resulting mainly from the rapid increase in aquaculture production in the last three decades (from 14.9 million tons in 1986 to 87.5 million tons in 2020), whereas, in the same period, global catches stabilized (from 86.9 million tons in 1986 to 90.3 million tons in 2020) after three decades of rapid increase (from ca. 20 million tons in 1950 to the referred ca. 80 million tons in 1986) [1]. Such statistics reflect the importance of seafood worldwide, either regarding their nutritional value (i.e., proteins, essential fatty acids or minerals) or socio-economic relevance due to their contribution to food security and economic improvement worldwide [2,3,4,5].

In this context of expanding fish production [1], seafood mislabeling has been escalating as high demand, high commercial value, limited resources and complex supply chains create opportunities for fraudulent activities [6,7]. Seafood can be mislabeled in terms of origin (harvesting location or production method) or in terms of species (since some species have higher reputation and economic value) [7,8]. For instance, the mislabeling of farmed fish as being wild-caught, for which consumers are willing to pay more, is well documented [7,9,10]. Therefore, the development of traceability systems around the world has been motivated by the growing public awareness about seafood mislabeling and its consequences (e.g., food safety risks or stock depletion), as well as the economic advantage regarding the consumers’ willingness to pay more for products for which the species and the origin have been traced [11,12,13,14,15]. As such, supported by chemometric tools [16,17], an array of biogeochemical signatures has been used to trace seafood provenance, among which stable isotopes, fatty acid profiles and elemental fingerprints are the most used [18]. For instance, the elemental fingerprints of fish muscle have been used to trace the production methods of various fish, such as the European seabass (*Dicentrarchus labrax*) [19,20], the Asian seabass (*Lates calcarifer*) [21] and several salmon species (*Oncorhynchus kisutch*, *Oncorhynchus kisutch* and *Salmo salar*) [22]. Despite the elemental signatures of calcified tissues, such as otoliths [23], having also been used in this scope, edible tissues (i.e., muscle) provide the advantage of simultaneously allowing us to trace the origin of seafood and evaluate food safety issues concerning the levels of potentially toxic elements. Additionally, fish are frequently sold as fillets, and in these cases, the only accessible sample matrix is the muscle.

The use of fish muscle elemental fingerprints as natural tags to trace fish origin is possible because, to a greater or lesser extent, elemental signatures reflect the chemical composition of the surrounding environment (i.e., water and sediments) and the fish diet [18], with the latter being the major contributor [24]. This elemental accumulation in fish muscle can be both an opportunity (as fish can represent a good source of essential elements [25]) or a food safety risk if they grow in contaminated areas or their diet contains high levels of potentially toxic elements [26]. While wild *S. aurata* is a carnivore (i.e., feeds on crustacea, molluscs, polychaetes and fish) with occasional periods of herbivory [27,28], the diet of farmed fish can be exclusively based on manufactured feeds, in the case of intensive aquaculture (e.g., in sea cages), as well as it can be a combination of both wild and manufactured feeding in semi-intensive aquaculture (e.g., in earth ponds) [29] or it can be similar to the wild, in the case of extensive aquaculture.

The gilthead seabream (*Sparus aurata*) is a good example of a species of high commercial interest from both wild-caught and farmed sources, especially in southern Europe and the Mediterranean basin [4,5]. In fact, *S. aurata* is one of the most important farmed fish in Europe in both value and volume [4], which has likely contributed to cases of mislabeling of *S. aurata* regarding catch and production methods [10]. Due to the need for the development of food safety and provenance assessments for this and other species, in the present study, the elemental fingerprints of *S. aurata* muscle from four different origins and from three production methods (wild-caught, aquaculture in earth ponds (semi-intensive) and sea-cage aquaculture (intensive)) were assessed, aiming to evaluate the following: (i) if the production method of this socio-economically important species can be traced using an edible muscle multi-elemental signature and (ii) the edible muscle levels of elements with regulatory thresholds (i.e., Cu, Hg, Pb and Zn).

## 2. Materials and Methods

### 2.1. Study Areas and Sample Collection

The samples of *Sparus aurata* were collected in two locations in the Northeast Atlantic and from three production methods as follows: in Olhão (Portugal Mainland, FAO fishing area of 27.9.a) from (i) wild (Owild), (ii) semi-intensive aquaculture in earth ponds (Opond), (iii) intensive sea-cage aquaculture (Ocage) and (iv) in the Madeira Island (FAO fishing area of 34.1.2) from intensive sea-cage aquaculture (Mcage) (Figure 1). Thirty specimens were collected per site and production method in November 2021, a collection period that was determined by the availability of all production sources.

The three production methods are characterized by different feeding regimes, namely, the natural diet of wild fish, the manufactured feeding of fish in intensive sea-cage aquaculture and the combined diet (i.e., natural diet complemented with manufactured feeds) of fish in semi-intensive aquaculture in earth ponds. After the collection, the specimens were transported fresh to the laboratory, where they were individually weighed (total weight in g) and measured (total length in cm). Then, they were dissected with plastic scissors and tweezers to collect muscle tissue from one fillet for elemental analysis. The dissected muscle tissues were stored at −80 °C until further analysis.

### 2.2. Elemental Analysis

After freeze-drying (at −50 °C, Telstar laboratory freeze dryer, Cryodos-45), the *S. aurata* muscles (the weight range was 0.213–0.279 g) were mineralized in Teflon reactors with 3.2 mL of HNO_3_ (70% *v*/*v*) through microwave digestion (Multiwave GO, Anton Paar GmbH, Graz, Austria) [30]. Gallium (certified reference material (CRM) for inductively coupled plasma spectrometry with purity >99.5%) was added to each sample mineralization product as an internal standard (the final concentration was 1 mg/L). On a siliconized quartz disk (BrukerNano, Berlin, Germany), 5 µL of each sample was applied and dried. The elemental compositions (As, Br, Ca, Cl, Cr, Cu, Fe, Hg, K, Mn, Na, Ni, P, Pb, Pr, Rb, S, Sb, Se, Sm, Sr, Ti, V, Y and Zn) were determined using total reflection X-ray fluorescence spectroscopy (TXRF S2 PICOFOX, Bruker, Berlin, Germany). For quality assurance and control (QA/QC), instrumental recalibration (gain correction, sensitivity analysis and multi-elemental standards) and analytical blanks were performed. The final elemental concentrations were determined by comparisons with the internal standard [17,31], and the recovery efficiency of each element was confirmed through the analysis of the reference materials (ERM-BB422 fish muscle) (Table 1). Considering the high extraction efficiency verified for the certified elements and the unavailability of CRM for fish muscle matrixes with a wider array of standards, using the abovementioned protocol, it can be assumed that the remaining analyzed elements have similar high extraction efficiencies.

### 2.3. Data and Statistical Analysis

To assess if the multielement profile of the *S. aurata* muscle can be used to trace its method of production, a random forest classifier was built using a leave-one-out cross-validation with multidimensional scaling ordination (MDS) of proximity scores being produced [32,33]. The random forest was built with 3000 classification trees to assure the stabilization of classification errors and with the number of variables randomly sampled as candidates at each split being the square root of p (p is the number of analyzed elements). Moreover, to test if the geographic variability of the elemental fingerprints of the *S. aurata* muscle impairs the allocation of the samples to their original production method, the samples of Madeira Island (Mcage–sea cage) were used as a test dataset in a second random forest model developed with samples from Olhão, representing the three production methods (i.e., Owild–wild, Opond–earth ponds and Ocage–sea cage). This approach was performed to replicate a real-case scenario where it would be necessary to trace the production method of *S. aurata* without reliable baseline information about its harvesting location. The elemental contribution to the model’s accuracy was evaluated through the Gini score provided by the random forest approach.

To compare the morphometric differences between the individuals collected at the considered sources, Kruskal–Wallis non-parametric tests with a Bonferroni correction were performed. The Spearman correlation coefficients between the morphometric variables (length and weight) and the element concentrations were calculated; the significant correlations were assessed. Then, for each element, the non-parametric Kruskal–Wallis post-hoc tests with the Bonferroni correction were used to evaluate if significant differences between the pairs of origins existed. Moreover, in the scope of the food safety assessment, the international regulatory or the recommended maximum concentrations in fish muscle were used to compare with the concentrations of potentially toxic elements in *S. aurata* muscle as follows: Cu (30 mg/kg), Hg (0.5 mg/kg), Pb (0.30 mg/kg) and Zn (40 mg/kg) [34,35].

Differences between the origins were considered significant at *p* ≤ 0.05. All statistical analyses were performed in R (v. 4.1.3), with corrplots, non-parametric Kruskal–Wallis tests, boxplots, random forest classifiers and chord diagrams being realized using the “corrplot”, “agricolae”, “ggplot2”, “randomForest” and “circlize” packages, respectively [33,36,37,38,39,40].

## 3. Results

### 3.1. Origin and Production Method Authentication

The random forest model based on the elemental fingerprints of *S. aurata* muscle from all four sampling origins presented an overall classification success of 90% (Table 2, Figure 2). The accuracy success ranged from 83.3% for Opond to 100% for Mcage (Table 2), with the few misclassifications all occurring among samples collected in the region of Olhão (i.e., Owild, Opond and Ocage), which are the origins sharing the highest mean proximity scores (Table 2, Figure 3). The elements that contributed the most to the accuracy of the random forest classification were as follows: Se and As, with Pb, Rb, Br, S and Mn also presenting substantial contributions (Figure 4).

The training model built with samples only from the collection site where all production methods are present (Olhão, i.e., Owild, Opond and Ocage) presented an overall cross-validation accuracy of 90% (Table 3). Using this model to classify the Madeira (Mcage) samples as a test dataset, it was possible to allocate correctly 27 out of 30 samples (90%) according to their production method, namely, sea-cage aquaculture (i.e., Ocage in Table 3). 

### 3.2. Morphometric Data, Total Elemental Fingerprinting and Food Safety

Both fish length and weight varied significantly among the sources, with significantly larger individuals obtained in Owild and significantly heavier fish from Owild and Mcage, whereas specimens from Ocage were smaller and lighter (Table 4). Positive and negative significant correlations were found between fish length or weight and the elements analyzed (Figure 5). These significant correlations were observed between As (positive) and Br (positive) and fish length, as well as between Br (positive), Ca (negative) and Ni (negative) and fish weight. Regarding the Spearman correlations between the elements, all elements showed at least one significant correlation with another, with V being the exception, with no significant correlations observed (Figure 5).

Some element concentrations (mg/Kg wet weight), such as As, Br, K, P, Pb and Se, showed clear variations in concentration among the origins, while others did not (Figure 6). Arsenic (Owild range = 0.983–15.439 mg/Kg), Br (Owild range = 1.309–9.830 mg/Kg) and Se (Owild range = 0.255–2.788 mg/Kg) presented significantly higher concentrations in the specimens from Owild, while significantly higher concentrations of K (Opond range = 5134.68–9624.66 mg/Kg) and P (Opond range = 1543.85–4045.39 mg/Kg) were measured in the specimens from Opond (Figure 6, Appendix A). Additionally, significantly higher Pb concentrations were observed in the specimens from Owild and Opond (Owild range = 0–2.703 mg/Kg, Opond range = 0.127–2.065 mg/ Kg) (Figure 6, Appendix A).

The concentrations of the elements with regulatory thresholds for human consumption ranged as follows: Cu (0.140–1.139 mg/Kg), Hg (0–0.506 mg/Kg), Pb (0–2.703 mg/Kg) and Zn (6.502–18.807 mg/Kg) (Figure 6, Appendix A). All of the *S. aurata* specimens presented concentrations below the recommended limits for Cu and Zn, while for Hg, only one specimen presented a concentration above the recommended threshold (Figure 6, Appendix A). Nonetheless, several *S. aurata* specimens from all of the origins presented Pb concentrations above the recommended threshold, with special emphasis on Owild and Opond, where more than 50% of the specimens presented Pb concentrations surpassing the maximum limit recommended (Figure 6, Appendix A).

## 4. Discussion

### 4.1. Sparus aurata Traceability

Consumers’ concerns about the sustainability of fisheries and the consequent pressure for improved labeling and traceability systems have resulted in the development of traceability tools targeting a wide range of economically and ecologically relevant seafood species [18,41,42]. Although variations in the elemental fingerprints of *S. aurata* muscles from different production methods have been previously reported [43,44,45,46], this marker has not been previously exploited to trace the production method of this commercially important fish species. As one of the most farmed fish species in Europe (of similar importance to seabass and outranked only by salmon and trout and by oysters and mussels in terms of volume but not value), this specific issue is of the utmost importance [4,5]. Elemental fingerprinting has been successfully used to determine the production method, with the allocation accuracy of the model from the present study (i.e., 90%) in line with the best results of those studies (Table 5) [19,21,22]. Some elements (Se, As, Pb, Rb, Br, S and Mn) varied substantially among the origins; thus, being the most important features for the allocation of the samples to their correct origin. This set of elements is considerably different from those groups considered as the most important for the determination of the provenance of other seafood, e.g., [17,31], suggesting that the most important elements for the classification are tissue- or species-specific. Moreover, the hypothesis that other biochemical signatures (i.e., fatty acids and isotopic profiles) of *S. aurata* muscles hold the potential to trace their method of production has already been advanced [47]. However, in this previous study, only two groups (farmed vs. wild) from different countries (Greece and Spain) were used, which does not allow us to resolve if the differences in biochemical signatures were due to the production method or the different geographic locations [47]. In the present study, this confounding factor (i.e., harvesting location) was eliminated since the samples of the three production methods were collected in the same (or nearby) geographic area (i.e., Olhão).

One of the key aims of developing traceability tools is to detect mislabeling or missing information on seafood production methods or harvesting locations [7]. Therefore, the present study evaluated if a model based only on samples from one location (in this case, Olhão in Mainland Portugal), representing three production methods (i.e., wild-caught, aquaculture in earth ponds and sea-cage aquaculture), would accurately predict the production method of samples from another location (in this case, sea-cage aquaculture in Madeira Island). The high accuracy obtained in this model, with 27 out of 30 specimens of Madeira Island being allocated to the correct production method, reinforces the potential of elemental fingerprinting of *S. aurata* muscle to trace their production method, regardless of the spatial variation associated with fish-harvesting locations [21,31]. Moreover, Davis et al. [16] reported that the large majority of seafood traceability studies used the same data to build and test the models, which can overestimate their accuracies. This was not the approach used in this model, where independent training and test datasets were used and high allocation success was attained, reinforcing the robustness of this methodology (i.e., elemental fingerprinting of fish muscle). Nevertheless, despite the high accuracy of the models presented here, to assure the success of fish traceability in future studies, the optimization of the traceability tools must be pursued by, for instance, using complementary biochemical signatures (e.g., stable isotopes) [18,20,22,41].

### 4.2. Morphometric Data, Elemental Fingerprinting and Food Safety

Several studies have explored the relationship between the size (weight and length) and the element concentrations in fish muscle, with some reporting negative correlations and others positive [48] and references therein, or even finding several significant correlations [49], while others found few or non-significant correlations [50]. A study comparing several elements in *S. aurata* muscle and its size found no significant correlations [51], which, in general, is in line with the results of the present study, where only two (out of 25) elements presented significant correlations with length or weight. The low relationship of morphometrics with the element concentrations indicates that diet and local environmental chemistry were the main factors driving the variation in the elemental fingerprint of *S. aurata* muscle, as previously described for fish in general [18,24].

Wild-caught (Owild) and earth-pond aquaculture (Opond) fish presented the highest concentrations of As, Br and Se, and K and P, respectively. Additionally, Owild and Opond presented the highest mean concentrations of Mn, Pb and S when compared with fish from sea-cage aquaculture (Ocage and Mcage). This is likely related to their feeding (natural for Owild and a combination of natural and manufactured for Opond) and the production site located just outside (Owild) and within (Opond) a coastal lagoon (Ria Formosa), which represents a risk of potential anthropogenic loadings. Indeed, the presence of high concentrations of As, Br and Se in Owild specimens indicates the high bioavailability of these elements in the food chain. Despite the lack of guidelines for total As concentration in seafood, special attention should be given to this element due to its potential high toxicity for humans, particularly in its inorganic forms [52].

In the present study, higher concentrations of total As were found in the muscle of wild *S. aurata* when compared with specimens from aquaculture, which was also described previously for fish from Portugal [53] and elsewhere [43,44,45,54]. This is likely a consequence of their natural diet, which is rich in crustaceans and, thus, presents high concentrations of this element [53,55]. However, as the non-toxic organic forms of As are usually the majority in seafood, when compared with the toxic inorganic forms, estimated to represent only up to 10% of the total [44,52,56], the consumption of wild *S. aurata* can be considered safe. Considering the percentage of inorganic forms (10%) and the recommended threshold of inorganic As in fish (2 mg/Kg wet weight) [57], the total As would have to exceed 20 mg/Kg to surpass the threshold, which was not observed in any specimen in this study (As maximum = 15.439 mg/Kg; Appendix A). The high concentrations of Br and Se observed in the wild specimens suggest the consumption of seaweeds that contain high concentrations of both elements [58,59]. The higher concentrations of essential elements (K and P) in Opond individuals are in line with the well-described nutrient enrichment observed in coastal lagoons, such as Ria Formosa, which is usually linked with runoffs from anthropogenic activities (e.g., agriculture) [60,61].

The elements with the recommended thresholds (Cu, Hg and Zn) presented concentrations in *S. aurata* muscle from all origins below the recommended limits; thus, not representing a food safety issue. Similarly, low values were found in other studies for Portuguese farmed *S. aurata* [62] and wild thornback rays collected along the Western Portuguese coast [31], suggesting that both the feeding of all production methods and the Portuguese oceanic waters present a low concentration of these elements. On the other hand, several specimens of all origins showed Pb concentrations above its recommended threshold, which aligns with concentrations found in stalked barnacle peduncle and octopus ink (also coastal species, such as *S. aurata*) from several locations on the Portuguese coast [17,63], indicating a high bioavailability of Pb in Portuguese oceanic waters. Nonetheless, the high concentrations of Pb can be related to the sampling season (i.e., autumn), which has previously been associated with the highest seasonal concentrations of Pb in *S. aurata* muscle in another geographical area, namely, the Mediterranean [64]. Regarding the relationship between the production methods and the Pb concentrations in fish muscle, previous studies reached inconsistent results, with some recording the highest concentrations in wild-caught [54,65] and others in farmed fish [44,45], which can explain the nonexistence of significant differences among Pb concentrations in Owild and Opond specimens in this study.

## 5. Conclusions

The present study investigated the potential of the elemental fingerprint of *S. aurata* muscle to trace fish production methods and also explored if the elements exceeded their recommended thresholds, which was only the case for Pb, with *S. aurata* muscle concentrations consistently above the established guidelines. These results contributed to the development of traceability tools and their applicability in real-case scenarios, as well as regarding food safety, to raise attention to the potentially toxic concentrations of Pb for human consumption in fish from all origins and the high concentrations of As in wild-caught *S. aurata*. The authenticity tools presented in this study will contribute to the fair valorization of fish and the food safety of consumers towards the production of origin certificates and will help authorities and wholesalers in the verification of provenance claims. Future work should focus on the combination of biogeochemical signatures to trace the production method of *S. aurata*, assessing the trade-off between the financial and time costs vs. potential accuracy increment through comparison with the use of only one biochemical signature. Further studies should also evaluate the use of alternative tissues (e.g., fin rays) that allow for different applications (e.g., their sampling allows the maintenance of fish integrity).

## Figures and Tables

**Figure 1 foods-11-03081-f001:**
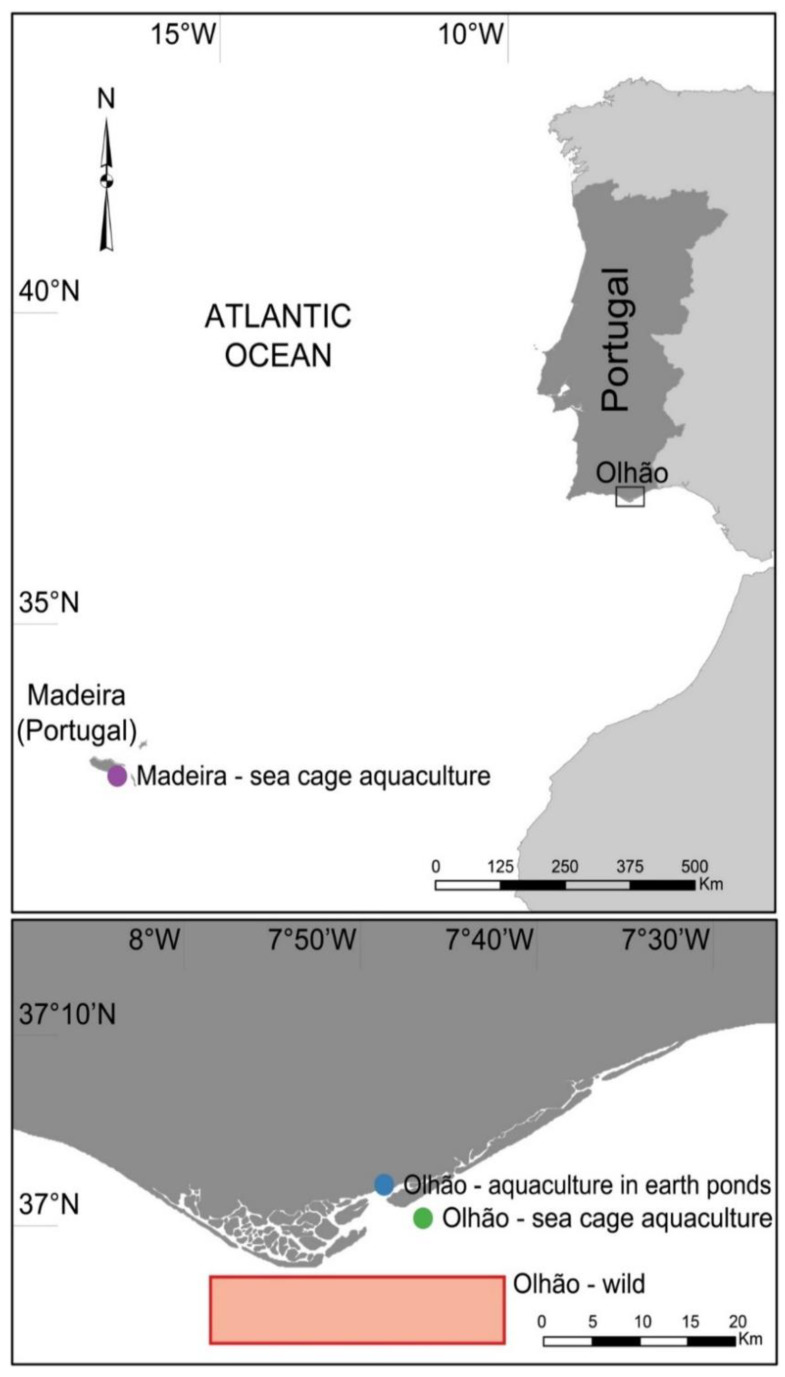
Location of wild and aquaculture collections of *Sparus aurata* in the southern Portuguese mainland coast (Olhão) and Madeira archipelago and from three fish production methods as follows: Olhão–wild (Owild), Olhão–aquaculture in earth ponds (Opond), Olhão–sea-cage aquaculture (Ocage) and Madeira–sea-cage aquaculture (Mcage).

**Figure 2 foods-11-03081-f002:**
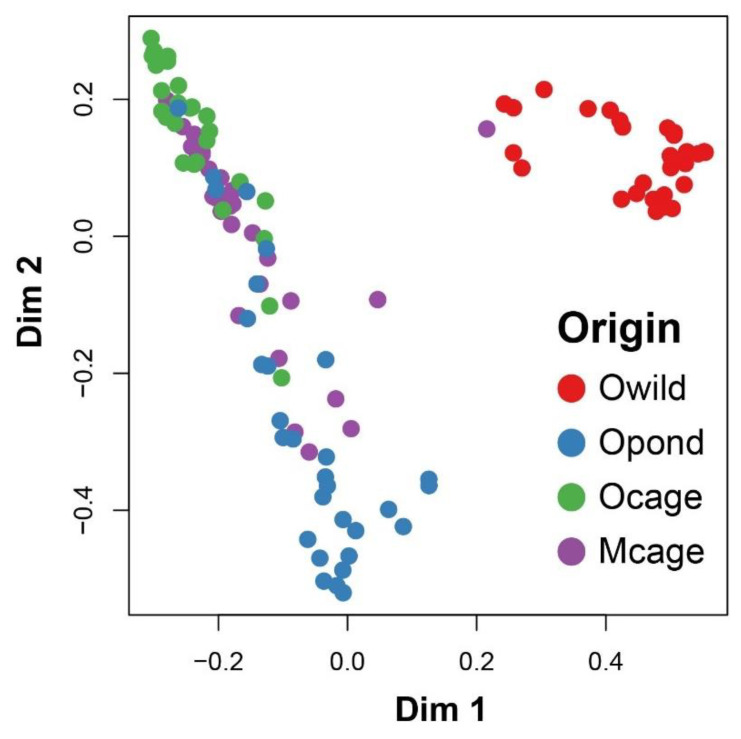
Multidimensional scaling (MDS) ordination of proximity scores from random forest classifications based on elemental fingerprints of *Sparus aurata* muscles collected at the following four sampling origins: Olhão–wild (Owild), Olhão–aquaculture in earth ponds (Opond), Olhão–sea-cage aquaculture (Ocage) and Madeira–sea-cage aquaculture (Mcage).

**Figure 3 foods-11-03081-f003:**
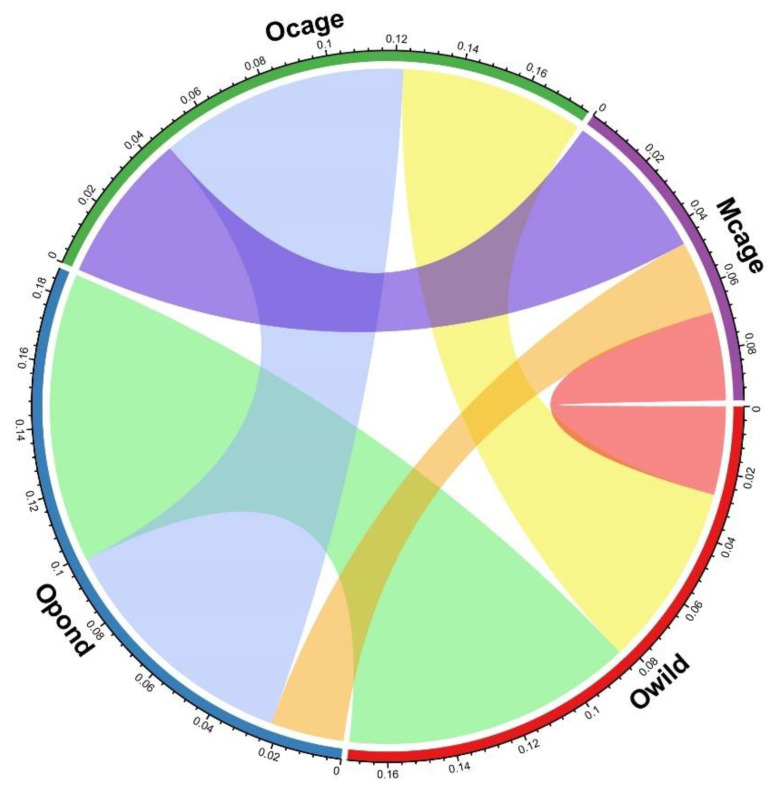
Chord diagrams based on average proximity scores from random forest classifications based on elemental fingerprints of *Sparus aurata* muscles collected at the four sampling origins. The wider alluvials represent higher average proximity scores between the paired sampling locations. Olhão–wild (Owild), Olhão–aquaculture in earth ponds (Opond), Olhão–sea-cage aquaculture (Ocage) and Madeira–sea-cage aquaculture (Mcage).

**Figure 4 foods-11-03081-f004:**
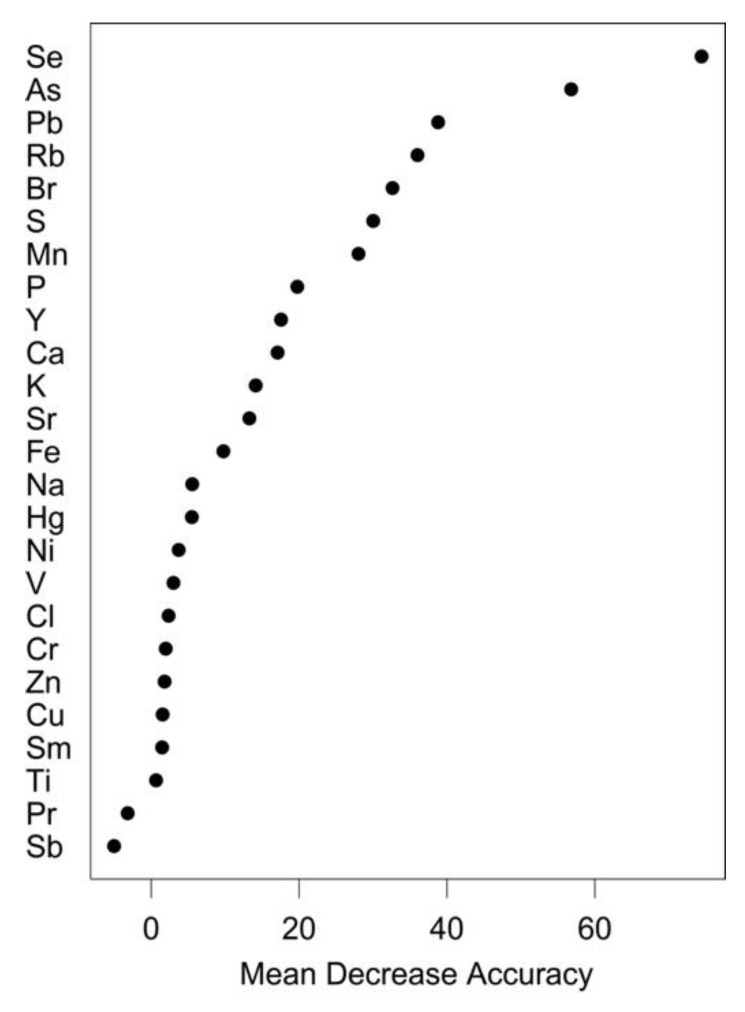
Importance of each element for the accuracy of the random forest classifications, according to the mean decrease in the accuracy measure, based on the elemental fingerprints of the muscle of *Sparus aurata* collected at the following four sampling origins. Olhão–wild (Owild), Olhão–aquaculture in earth ponds (Opond), Olhão–sea-cage aquaculture (Ocage), Madeira–sea-cage aquaculture (Mcage).

**Figure 5 foods-11-03081-f005:**
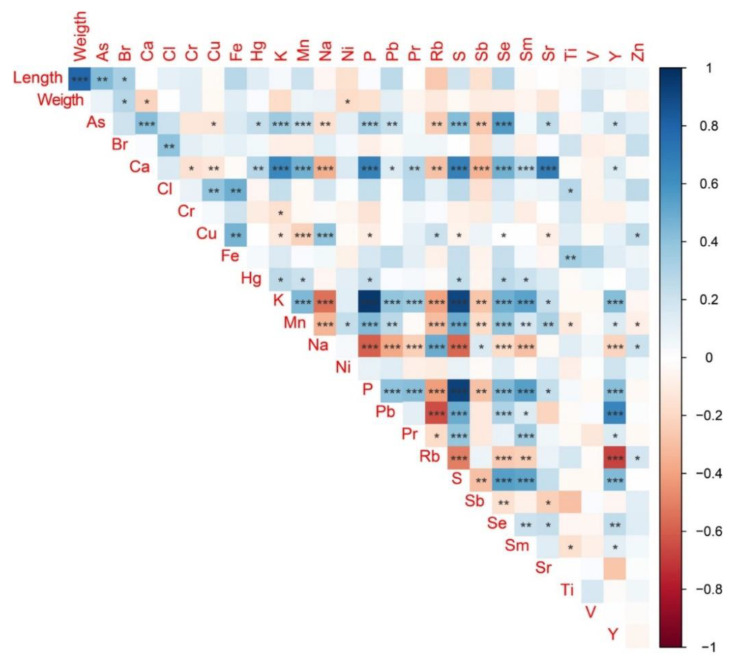
Spearman correlation matrix (ρ) between the morphometric variables (fish total length and weight) and the elemental concentrations (mg/Kg) detected in the *Sparus aurata* muscle (the asterisks denote significant correlations at *p* < 0.05 *, *p* < 0.01 ** and *p* < 0.001 ***).

**Figure 6 foods-11-03081-f006:**
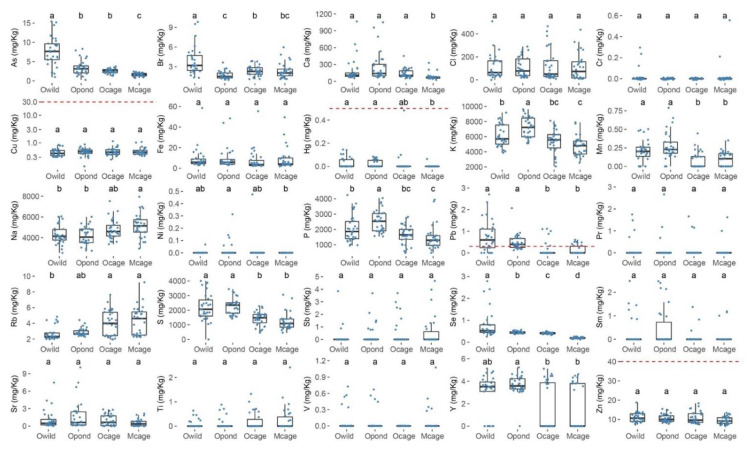
Elemental concentrations (mg/Kg) in the *S. aurata* muscle (wet weight) of the individuals collected at the following four sampling origins: Olhão–wild (Owild), Olhão–aquaculture in earth ponds (Opond), Olhão–sea-cage aquaculture (Ocage) and Madeira–sea-cage aquaculture (Mcage). The dotted red lines represent the safety threshold, according to the international regulatory authorities. Different statistical letters (a, b, c, and d) denote significant differences between the sampling sites at *p* < 0.05.

**Table 1 foods-11-03081-t001:** Fish muscle (ERM-BB422) certified and analyzed elemental values, uncertainty (mg/Kg) and calculated extraction efficiencies (average ± standard deviation, N = 5).

Element	Certified Value	Uncertainty	Measured Value	Extraction Efficiency (%)
Cr	0.73	0.22	0.67 ± 0.10	91.3 ± 5.3
Mn	4.88	0.24	3.35 ± 0.10	68.6 ± 1.9
Fe	161.0	8.0	198.03 ± 0.67	123.0 ± 0.4
Ni	0.69	0.15	0.78 ± 0.05	113.1 ± 5.8
Cu	5.98	0.27	7.10 ± 0.07	118.8 ± 1.0
Zn	71.0	4.0	73.29 ± 0.28	103.2 ± 0.4
As	6.7	0.4	7.25 ± 0.07	108.2 ± 0.9
Se	1.62	0.12	1.51 ± 0.03	93.3 ± 1.9
Rb	2.46	0.16	2.45 ± 0.05	99.6 ± 1.9
Sr	19.0	0.0	18.55 ± 0.34	97.6 ± 1.8
Cd	0.336	0.025	0.32 ± 0.02	96.6 ± 4.5
Pb	2.18	0.18	2.47 ± 0.05	113.3 ± 2.1

**Table 2 foods-11-03081-t002:** Classification accuracy (by production method) of the random forest classification based on the elemental fingerprints of *Sparus aurata* collected at the four sampling origins as follows: Olhão–wild (Owild), Olhão–aquaculture in earth ponds (Opond), Olhão–sea-cage aquaculture (Ocage) and Madeira–sea-cage aquaculture (Mcage).

Original Group	Classified Group				%Correct
	Owild	Opond	Ocage	Mcage	
Owild	26	2	2	0	86.7
Opond	3	25	2	0	83.3
Ocage	0	3	27	0	90
Mcage	0	0	0	30	100
Total					90

**Table 3 foods-11-03081-t003:** Classification accuracy (by production method) of random forest classifications based on the elemental fingerprints of *Sparus aurata* collected at Olhão (model training), and the allocation of the samples from Madeira Island in the origins of the training model (test). Olhão–wild (Owild), Olhão–aquaculture in earth ponds (Opond), Olhão–sea-cage aquaculture (Ocage) and Madeira–sea-cage aquaculture (Mcage).

	Original Group	Classified Group			%Correct
Model Training		Owild	Opond	Ocage	
	Owild	29	1	0	96.7
	Opond	3	25	2	83.3
	Ocage	0	3	27	90
	Total				90
Test	Predicted (Mcage)	0	3	27	

**Table 4 foods-11-03081-t004:** Total length and weight of *Sparus aurata* individuals (N = 120, 30 per origin) collected at the following four sampling origins: Olhão–wild (Owild), Olhão–aquaculture in earth ponds (Opond), Olhão–sea-cage aquaculture (Ocage), Madeira–sea-cage aquaculture (Mcage). Different letters (a, b, and c) denote significant differences between the sampling sites at *p* < 0.05.

	Length	Weight
Owild	33.75 ± 1.16 ^a^	563.04 ± 44.72 ^a^
Opond	28.18 ± 1.34 ^b^	342.68 ± 33.70 ^b^
Ocage	27.07 ± 1.00 ^c^	299.70 ± 38.70 ^c^
Mcage	28.42 ± 0.99 ^b^	534.93 ± 51.67 ^a^

**Table 5 foods-11-03081-t005:** Production method traceability studies based only on the elemental fingerprinting of fish muscle.

Species	Allocation Success (%)	Reference
Gilthead seabream (*Sparus aurata*)	90	This study
Asian seabass (*Lates calcarifer*)	72	[21]
European seabass (*Dicentrarchus labrax*)	56–79	[19]
Salmon species (*Oncorhynchus kisutch*, *Oncorhynchus kisutch*, *Salmo salar*)	56–92	[22]

## Data Availability

Data available upon reasonable request.

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
