# Peer review of "Elemental Fingerprinting of Wild and Farmed Fish Muscle to Authenticate and Validate Production Method"

_foods, 2022, doi:10.3390/foods11193081_

Round 1

Reviewer 1 Report

The article deals with the potential of the elemental fingerprint of S. aurata muscle to trace fish production methods and explored if elements exceeded recommended thresholds. The study presented scientific relevance to the field of food safety. There are some basic flaws, however, which must be fixed, and the Authors are invited to revise accordingly. Major comments:

Abstract

Authors should summarize the article's main findings, for example, what value of concentrations of Cu, Hg, Pb, and Zn in the manuscript?

Introduction

Page 1: lines 38-40: References were not up to the mark, try to give the latest for the past 2 years.

Materials and methods

2.2. Elemental analysis

Page 4: line 115: Authors should report the grade purity of internal standard.

Table 1: Why is extraction efficiency higher than 100%?

As was calculated extraction efficiency?

Results

The tables and Figures were adequate, maybe a comparative table can be given.

Author Response

Reviewer 1

The article deals with the potential of the elemental fingerprint of S. aurata muscle to trace fish production methods and explored if elements exceeded recommended thresholds. The study presented scientific relevance to the field of food safety. There are some basic flaws, however, which must be fixed, and the Authors are invited to revise accordingly. Major comments:

 Thank you for your insight and comments.

Abstract

Authors should summarize the article's main findings, for example, what value of concentrations of Cu, Hg, Pb, and Zn in the manuscript?

Response: Corrected as suggested. This information was added through modification of a sentence that now reads as: “The concentration of the regulated elements ranged as follows: Cu (0.140-1.139 mg/Kg), Hg (0-0.506 mg/Kg), Pb (0-2.703 mg/Kg) and Zn (6.502-18.807 mg/Kg),...”.

Introduction

Page 1: lines 38-40: References were not up to the mark, try to give the latest for the past 2 years.

Response: Corrected as suggested. The reference FAO (2020) was replaced by FAO (2022), with the information about fish trade being updated in the manuscript.

Materials and methods

2.2. Elemental analysis

Page 4: line 115: Authors should report the grade purity of internal standard.

Response: Corrected as suggested. This information was added.

Table 1: Why is extraction efficiency higher than 100%?

As was calculated extraction efficiency?

Response: The extraction efficiency was calculated through the following formula: Extraction efficiency = (measured value of element x in the reference material / Certified Value of element x in the reference material) *100. Considering that each element has a certified value ± an uncertainty value it is very common to have values that can surpass the 100%, and this issue is recognized as a common feature and included in all CRM guidelines for all materials in the market.

Results

The tables and Figures were adequate, maybe a comparative table can be given.

Response: As per the Reviewers’ comment, a comparative table with the concentrations range of each element by origin was already provided as supplementary material (Table S1), but not sure if this is what the referee asks for.

Reviewer 2 Report

GENERAL STATEMENT

The manuscript is devoted to to the problem of authenticating the origin and the production method of samples of Sea bream by exploiting the potential of their multi-elemental profile and taking advantage from advanced multivariate statistics. Overall, the paper is very well written and adds some interesting information to the already available data confirming the suitability of using such an analytical approach to assist traceability within the fishery sector and, at the same time, to assess the safety related to presence of toxic metals. Figures and Tables are clear, well formatted, and concisely summarize the results achieved. The topic is well discussed, and it is of particular interest to a large audience of readers working in food safety and quality sciences and analytical chemistry. 

ABSTRACT

1) L. 32-34: Considering that the main topic of this work was to verify the farming system of S. aurata and considering that according to current legislation, only the production method (wild vs farmed) is a mandatory information to be provided within the seafood label, I would not stress enough the problem of fish fraud within the abstract (L. 34). I’d rather focus just on traceability.

INTRUDUCTION

2) L. 38- 39: ‘last decades’ is repeated within the same sentence. Please, rephrase.

3) L. 39-40 and L. 41: Is this the most recent information about annual production?

4) L. 90-91: A general curiosity: considering that one of the purposes was to assess the safety of S. aurata samples in relation to their contamination with regulated heavy metals, why Cd was not measured?

MATERIALS AND METHODS

5) L. 94: Please, use Italic to identify S. aurata here and throughout the text.

6) L. 94: It would be interesting to know if the fishing area from which wild and sea-cage samples from Madeira were collected was close to the coast (i.e., the FAO fishing area 27.IX.a) or offshore (i.e., the FAO fishing area 27.IX.b).

7) L.95: ‘four’ production method (including wild-caught fish) instead of ‘three’.

8) L. 105: Which part of muscle tissue was specifically used for multi-elemental analysis? One of both fillets?

9) L. 112: Add some information about lyophilization (time-temperature and apparatus)

10) L. 113-115: Volume and concentration of HNO3, volume of Gallium solution added to samples and the weight of sample which was mineralized should be added.

11) L. 123 and Table 1: ERM-BB422 Fish Muscle is certified only for 12 elements, while 25 elements were quantified by the authors. Hence, some doubts about the accuracy of the analytical method in quantifying 13 out 25 elements are present. In general, more than one CRM should be analyzed so as to cover as many elements as possible. Please, explain such a choice.

RESULTS

12) Why Random Forest was chosen as machine learning algorithm to authenticate samples by production methods over other well-known and easiest-to-implement algorithms (e.g., classical discriminant analysis or soft-independent modelling of class analogy)?

13) L. 130: Please, provide a more detailed information about the parameters employed to build the Random Forest classifier to avoid overfitting (I mean, for example, the maximum number of three employed, the maximum depth of the three structures, etc.).

14) L 127-138: It should be better remarked in the text that two different Random Forest classifiers were built (one including the four method of production and one including only 3 method of productions, Table 2 and 3, respectively) since it is not immediate to understand.  

15) L. 148: Maximum limits of Cu and Zn in fish products are not included in the European Regulation 1881/2006. The European Commission Regulation 149/2008 set Levels of 30 mg/kg copper (as pesticide residue) in terrestrial food but not aquatic food, hence I would like to ask the authors to revise this part.

16) Multidimensional scaling (MDS) ordination of proximity scores plotted in Figure 2 clearly show one single Mcage samples clustering together with Owild samples. Didi the authors check for labeling errors during sample preparation? Which characteristics in terms of elemental profile did this sample show to cluster with Owild samples?

DISCUSSION

17) An attempt to explain the reason underlying the high discriminatory significance of Se, As, Pb, Rb, Br, S, and Mn should be provided in the Discussion section, so as to clarify the results already reported in Section 3.1.

CONCLUSIONS

18) I would suggest including the practical utility of the method and some suggestions for its implementation within the fisheries traceability system.

Author Response

Reviewer 2

GENERAL STATEMENT

The manuscript is devoted to to the problem of authenticating the origin and the production method of samples of Sea bream by exploiting the potential of their multi-elemental profile and taking advantage from advanced multivariate statistics. Overall, the paper is very well written and adds some interesting information to the already available data confirming the suitability of using such an analytical approach to assist traceability within the fishery sector and, at the same time, to assess the safety related to presence of toxic metals. Figures and Tables are clear, well formatted, and concisely summarize the results achieved. The topic is well discussed, and it is of particular interest to a large audience of readers working in food safety and quality sciences and analytical chemistry. 

 Thank you for your insight and comments.

ABSTRACT

1) L. 32-34: Considering that the main topic of this work was to verify the farming system of S. aurata and considering that according to current legislation, only the production method (wild vs farmed) is a mandatory information to be provided within the seafood label, I would not stress enough the problem of fish fraud within the abstract (L. 34). I’d rather focus just on traceability.

 Response: Corrected as suggested. The part regarding fish fraud was deleted, hence this sentence now reads as: “The present findings contribute to establish elemental fingerprinting as a reliable tool to trace fish production methods and underpin seafood authentication.”.  

INTRUDUCTION

2) L. 38- 39: ‘last decades’ is repeated within the same sentence. Please, rephrase.

Response: Corrected as suggested. This sentence now reads as: “The global fish trade has been steadily growing over the years, resulting mainly from the rapid increase in aquaculture production in the last three decades”.

3) L. 39-40 and L. 41: Is this the most recent information about annual production?

Response: The Reviewer raised a relevant question as this is not the most recent information. Therefore, the reference FAO (2020) was replaced by FAO (2022), with the information regarding fish trade being updated in the manuscript.

4) L. 90-91: A general curiosity: considering that one of the purposes was to assess the safety of S. aurata samples in relation to their contamination with regulated heavy metals, why Cd was not measured?

Response: The method used to analyze the elemental composition of S. aurata muscle (i.e., TXRF) perform an untargeted analysis. Therefore, Cd was not included in this analysis because the concentrations of Cd in S. aurata muscle were below the detection limit of TXRF.

MATERIALS AND METHODS

5) L. 94: Please, use Italic to identify S. aurata here and throughout the text.

Response: Corrected as suggested.

6) L. 94: It would be interesting to know if the fishing area from which wild and sea-cage samples from Madeira were collected was close to the coast (i.e., the FAO fishing area 27.IX.a) or offshore (i.e., the FAO fishing area 27.IX.b).

Response: This information was added to the manuscript, as suggested by Reviewer 2.

7) L.95: ‘four’ production method (including wild-caught fish) instead of ‘three’.

Response: We understand the comment, but that is not entirely correct. In this paper, we evaluated the elemental fingerprints of fish from four origins, embracing however only three production methods (wild-caught, earth ponds and sea cages) from two locations. The two origins shared a production method (i.e., sea cages). In sum, the origins evaluated were: Olhão - wild (Owild), Olhão - aquaculture in earth ponds (Opond), Olhão - sea cage aquaculture (Ocage), Madeira - sea cage aquaculture (Mcage).

8) L. 105: Which part of muscle tissue was specifically used for multi-elemental analysis? One of both fillets?

Response: The Reviewer raised an important question. Therefore, this information was added to the manuscript.

9) L. 112: Add some information about lyophilization (time-temperature and apparatus)

Response: As suggested by Reviewer, information about lyophilization was provided.

10) L. 113-115: Volume and concentration of HNO3, volume of Gallium solution added to samples and the weight of sample which was mineralized should be added.

Response: The Reviewer raised an important question regarding the extraction volume and therefore, this information was added to the manuscript. Regarding the volume of Gallium, it depends on the volume of mineralized product used from the 3.2 mL extraction volume, since TXRF only requires very small volumes for analysis, and thus not all 3.2 mL are used for analysis. The relevant information concerning the Ga final concentration is presented in the manuscript, which will allow reproducibility if others want to use a different volume.

11) L. 123 and Table 1: ERM-BB422 Fish Muscle is certified only for 12 elements, while 25 elements were quantified by the authors. Hence, some doubts about the accuracy of the analytical method in quantifying 13 out 25 elements are present. In general, more than one CRM should be analyzed so as to cover as many elements as possible. Please, explain such a choice.

Response:  In fact, we partially agree with Reviewer as it would be ideal if the elements detected in the samples had certified values. However, even if we had used two certified materials it would be unlikely to cover the 25 elements found in S. aurata muscle. Finally, as the TXRF is an untargeted method, it is very difficult to cover every possible combination of elements detected. Moroever the CRM used should be as close as possible in terms of matrix composition to the samples to be analyzed in order to evaluate if the extraction procedure would be suitable for a certain matrix. Although we have several other CRM available (e.g. algae, plants, sediments) with other elemental compositions their matrix composition is very different from fish muscle and thus would not allow to certify if the extraction procedure is or not efficient for fish muscle samples. Moreover and considering the extraction efficiency of the certified elements presented in the present work, it is fair to assume that other elements would be equally extracted with similar efficiency, being this a common practice when surveying a large array of elements, that are also common to be absent in CRMs.

RESULTS

12) Why Random Forest was chosen as machine learning algorithm to authenticate samples by production methods over other well-known and easiest-to-implement algorithms (e.g., classical discriminant analysis or soft-independent modelling of class analogy)?

Response: Random Forest was chosen as this is a statistical method commonly used to determine the provenance of seafood based on its elemental fingerprints (e.g., Bennion et al., 2019; Morrison et al. 2019; Bernardo et al., 2022; Mamede et al., 2022). The authors consider that Random Forest classification is not complicated to implement and is a very accurate statistical method, which is supported by the high percentage of samples classified in the studies referenced above and in the present study (90%).

  1. Morrison, L.; Bennion, M.; Gill, S.; Graham, C.T. Spatio-temporal trace element fingerprinting of king scallops (Pecten maximus) reveals harvesting period and location. Sci. Total Environ. 2019, 697, 134121, doi:10.1016/j.scitotenv.2019.134121.
  2. Duarte, B.; Mamede, R.; Duarte, I.A.; Caçador, I.; Tanner, S.E.; Silva, M.; Jacinto, D.; Cruz, T.; Fonseca, V.F. Elemental Chemometrics as Tools to Depict Stalked Barnacle (Pollicipes pollicipes) Harvest Locations and Food Safety. 2022, 1–16, doi:doi.org/10.3390/molecules27041298.
  3. Bennion, M.; Morrison, L.; Brophy, D.; Carlsson, J.; Abrahantes, J.C.; Graham, C.T. Trace element fingerprinting of blue mussel (Mytilus edulis) shells and soft tissues successfully reveals harvesting locations. Sci. Total Environ. 2019, 685, 50–58, doi:10.1016/j.scitotenv.2019.05.233.
  4. Mamede, R.; Ricardo, F.; Gonçalves, D.; Ferreira da Silva, E.; Patinha, C.; Calado, R. Assessing the use of surrogate species for a more cost-effective traceability of geographic origin using elemental fingerprints of bivalve shells. Ecol. Indic. 2021, 130, 108065, doi:10.1016/j.ecolind.2021.108065.

13) L. 130: Please, provide a more detailed information about the parameters employed to build the Random Forest classifier to avoid overfitting (I mean, for example, the maximum number of three employed, the maximum depth of the three structures, etc.).

Response: In our view, overfitting is not an issue when using Random Forest classifiers, as argued by Breiman (2001) when giving some theoretical background of Random Forests : “Use of the Strong Law of Large Numbers shows that they always converge so that overfitting is not a problem”. As such, the depth of the tree structure does not influence the overfitting of Random Forest, which is well explained in this data science website: https://towardsdatascience.com/one-common-misconception-about-random-forest-and-overfitting-47cae2e2c23b. The non-overfitting of our models was confirmed when an independent training and test datasets were used and, even though, a high allocation success of the respective production method was attained (27 in 30 samples). Nonetheless, information about the tunning of some Random Forest parameters (e.g., maximum number od trees) was provided.

Breiman, L. Random forests. Mach. Learn. 2001, 45, 5–32, doi:10.1201/9780367816377-11.

14) L 127-138: It should be better remarked in the text that two different Random Forest classifiers were built (one including the four method of production and one including only 3 method of productions, Table 2 and 3, respectively) since it is not immediate to understand.  

Response: Corrected as suggested. This now read as: “the samples of Madeira Island (Mcage – sea cage) were used as a test dataset in a second Random Forest model developed...”.

15) L. 148: Maximum limits of Cu and Zn in fish products are not included in the European Regulation 1881/2006. The European Commission Regulation 149/2008 set Levels of 30 mg/kg copper (as pesticide residue) in terrestrial food but not aquatic food, hence I would like to ask the authors to revise this part.

Response: In fact, we acknowledged that maximum limits of Cu and Zn in fish are not included in the European Regulation 1881/2006. However, we used other document (FAO/WHO 1989, citation n. 34 in the manuscript) where maximum limits for Cu and Zn in fish were set.

FAO/WHO Evaluation of Certain Food Additives and the Contaminants Mercury, Lead and Cadmium. WHO Technical Report Series No. 505. 1989.

16) Multidimensional scaling (MDS) ordination of proximity scores plotted in Figure 2 clearly show one single Mcage samples clustering together with Owild samples. Didi the authors check for labeling errors during sample preparation? Which characteristics in terms of elemental profile did this sample show to cluster with Owild samples?

Result: The authors were very careful regarding the labelling of samples and, thus, we are pretty sure that this was not due a labelling error. We think that this is likely a misrepresentation of the distance among the elemental composition of the samples, since the MDS is a 2D representation of a multidimensional data cloud. This is supported by the fact that all samples of Mcage were correctly allocated to their origin.

DISCUSSION

17) An attempt to explain the reason underlying the high discriminatory significance of Se, As, Pb, Rb, Br, S, and Mn should be provided in the Discussion section, so as to clarify the results already reported in Section 3.1.

Response: Corrected as suggested. An explanation regarding the reason why these elements presented such discriminatory power is now included in the revised version of the manuscript. This sentence reads as: “Some elements (Se, As, Pb, Rb, Br, S and Mn) varied substantially among origins, thus being the most important features to the allocation of samples to their correct origin. This set of elements is considerably different from those groups considered as the most important to the determination of provenance of other seafood [e.g., 17, 31], suggesting that the most important elements for the classification are tissue/species specific.” (lines 256-261).

CONCLUSIONS

18) I would suggest including the practical utility of the method and some suggestions for its implementation within the fisheries traceability system.

Response: Corrected as suggested. The practical utility and some suggestion regarding the implementation of such authenticity tool were included in conclusions. This sentence reads as: ”The authenticity tools presented in this study will contribute to the fair valorisation of fish and food safety of consumers towards the production of origin certificates and helping authorities/wholesalers in the verification of provenance claims”.

Reviewer 3 Report

I have gone through the manuscript. The manuscript is well written, and I would like to suggest minor revision. The results of this study will contribute to the development of traceability systems of global fish production and aquaculture.

Line 98: Why did you collect the specimens in November? Do you believe the fingerprinting will be successful for specimens collected in other months? Please describe the reasons or consider further investigation.

Line 158: “(83.3%)” should be deleted.

Line 186-187: I don't get the meaning of this sentence. Additionally, in Figure 3, you should explain to the readers in more detail how to interpret the chord diagram.

Line 200-202: Since significant correlations between elements are shown except for V, is it possible to narrow down the number of elements and make a prediction? (e.g., Se or As that contributed most to the accuracy of the Random Forest classification, and the other additional a few elements only.) If it is possible, rapid and simplified methods might come true. Please consider further investigation.

Author Response

Reviewer 3

I have gone through the manuscript. The manuscript is well written, and I would like to suggest minor revision. The results of this study will contribute to the development of traceability systems of global fish production and aquaculture.

Line 98: Why did you collect the specimens in November? Do you believe the fingerprinting will be successful for specimens collected in other months? Please describe the reasons or consider further investigation.

Response: Collection was determined by the availability of all production sources, from wild-caught to farmed S. aurata. This information was added to the manuscript. Based on our experience in determining the provenance of seafood (including fish), we have observed some variations among the elemental fingerprints of provenances, yet in this case we believe that the elemental fingerprints of fish from different production methods will be differentiable independently of the time of the year. Still, we are working with the temporal variability of elemental fingerprints of fish in our investigations outside the scope of this manuscript.  

Line 158: “(83.3%)” should be deleted.

Response: Corrected as suggested.

Line 186-187: I don't get the meaning of this sentence. Additionally, in Figure 3, you should explain to the readers in more detail how to interpret the chord diagram.

Response: In this sentence we tried to explain why the majority of Mcage samples were allocated to Ocage, even referencing a figure regarding a different Random Forest Model (Figure 3). We now understand that this is not easily understandable and, hence, it was deleted.

Moreover, the following information about the interpretation of the chord diagram was added in the Figure caption: “Wider alluvials represent higher average proximity scores between paired sampling locations.”.

Line 200-202: Since significant correlations between elements are shown except for V, is it possible to narrow down the number of elements and make a prediction? (e.g., Se or As that contributed most to the accuracy of the Random Forest classification, and the other additional a few elements only.) If it is possible, rapid and simplified methods might come true. Please consider further investigation.

Response: In fact, we tested some methods to select variables based on their importance to the classification accuracy (e.g., Boruta algorithm (Kursa & Rudnicki, 2010)). However, none of the selections gave a classification accuracy higher than obtained using all variables. Therefore, we choose the more straightforward approach of using all elements in the Random Forest classification.

Kursa, M. B., & Rudnicki, W. R. (2010). Feature selection with the boruta package. Journal of Statistical Software, 36(11), 1–13. https://doi.org/10.18637/jss.v036.i11